# *In Vitro* Digestion of Chestnut and Quebracho Tannin Extracts: Antimicrobial Effect, Antioxidant Capacity and Cytomodulatory Activity in Swine Intestinal IPEC-J2 Cells

**DOI:** 10.3390/ani10020195

**Published:** 2020-01-23

**Authors:** Serena Reggi, Carlotta Giromini, Matteo Dell’Anno, Antonella Baldi, Raffaella Rebucci, Luciana Rossi

**Affiliations:** Dipartimento di Scienze Veterinarie per la Salute, la Produzione Animale e la Sicurezza Alimentare Carlo Cantoni, Università Degli Studi di Milano, Via Trentacoste 2, 20134 Milano, Italy; serena.reggi@unimi.it (S.R.); matteo.dellanno@unimi.it (M.D.); antonella.baldi@unimi.it (A.B.); raffaella.rebucci@unimi.it (R.R.); luciana.rossi@unimi.it (L.R.)

**Keywords:** plant extracts, tannin, *in vitro* digestion, growth inhibition, enterotoxigenic *E. coli*, antimicrobial activity, antioxidant capacity, IPEC-J2 intestinal cell

## Abstract

**Simple Summary:**

*Castanea sativa Mill.* (Fagaceae) is the predominant sweet chestnut tree in Europe. Despite the significant economic value of chestnuts as sources of food and wood, the high content of tannin also increases the value of sweet chestnut trees. Quebracho trees (*Schinopsis* spp., family *Anacardiaceae*) grow mainly in Argentina and Paraguay. Quebracho extract obtained from *Schinopsis* spp. contains 15% to 21% pure tannin. Tannins extracted from these plants have been applied in intensive swine farms due to their ability to improve animal performance and health. However, there are contrasting results regarding the bio-accessibility of chestnut and quebracho and their relative antioxidant activity and growth-rate reducing ability on *E. coli*, which ultimately affect their benefits in terms of intestinal health and animal production. Our results demonstrate that chestnut and quebracho exert a growth inhibitory activity against Enterotoxigenic *E. coli* (ETEC) species and antioxidant capacity directly, without extraction and after *in vitro* digestion. Our findings not only suggest that the combined use of chestnut and quebracho can maximize their functional effects, but also that an appropriate dosage of tannins may be key in terms of their effect on bacteria and cells.

**Abstract:**

Quebracho (Qu) and chestnut (Ch) are natural sources of tannins and they are currently used in animal nutrition as feed ingredients. However, to date the bio-accessibility, antimicrobial, antioxidant, and intestinal epithelial cell stimulatory doses of Qu and Ch have not been determined. Our study investigates the antioxidant and *E. coli* F4+ and F18+ growth inhibitory activity of Qu, Ch, and their combinations after solubilization in water (to evaluate the already bio-accessible molecules) and after simulated gastro-intestinal digestion *in vitro*. The effect of an *in vitro* digested Ch and Qu combination was also tested on intestinal epithelial IPEC-J2 cells experimentally stressed with hydrogen peroxide (H_2_O_2_) and Dextran Sodium Sulfate (DSS). The results showed that undigested Qu and Ch alone, and in combination, exerted a valuable antioxidant capacity and *E. coli* F4+ and F18+ growth inhibitory activity. The concentration of 1200 µg/mL exhibited the highest *E. coli* growth inhibitory activity for all the samples tested. In addition, after *in vitro* digestion, Qu and Qu50%–Ch50% maintained *E. coli* growth inhibitory activity and a modest antioxidant capacity. Three hours pre-treatment with *in vitro* digested Qu50%–Ch50% counteracted the H_2_O_2_ and DSS experimentally-induced stress in the intestinal IPEC-J2 cells. Ch and Qu tannin extracts, particularly when combined, may exert *E. coli* F4+ and F18+ growth inhibitory activity and valuable antioxidant and cell viability modulation activities.

## 1. Introduction

Plant tannins are water-soluble polyphenolic compounds of a variable molecular weight, which are abundant in nature [1]. They can be classified into condensed (molecular weight: 1000–20,000) and hydrolyzable groups (molecular weight: 500–3000) [2,3]. They have different nutritional significance and adverse effects. Chestnut (Ch, *Castanea sativa mill.*) and quebracho (Qu, *Schinopsis* spp.) tannin extracts have been used for over ten years in animal feeding [4,5,6]. Although tannins can interfere with the digestion of nutrients as they bind proteins and delay the absorption of sugar and lipids, several studies have reported that the addition of Qu and Ch to animal feed improved the growth performance and health in both ruminants and monogastric animals [4,5,7]. 

Although Ch and Qu tannins have been studied in both weaned and fattening piglets in terms of their antimicrobial activities [8,9], there is still no consensus on the appropriate Ch and Qu tannin dose that maximizes the beneficial effect and minimizes the anti-nutritional value of tannins. Moreover, even though Qu and Ch have shown significant biological properties *in vivo*, little is known about the bio-accessibility and bioavailability after digestion and the biological effects of such compounds used alone and in combination. Due to their chemical composition, they can exert antimicrobial, antiviral, antioxidant, and antimutagenic effects locally in the intestine as unadsorbable complex structures [10]. In this scenario, *in vitro* digestion models for nutrient evaluation are important for studying both the physiology of certain segments of the digestive tract and the digestive and bioactive characteristics of tannins. In pig livestock, Enterotoxigenic *E. coli* (ETEC) is the most important pathotype causing both neonatal and post-weaning diarrhea (PWD), which are responsible for significant economic losses worldwide and are the most common reason for the prescription of antimicrobials in intensive systems [11,12]. The proliferation of ETEC strains in the gut and their pathogenicity can be influenced by the expression of fimbrial adhesins, which bind to specific receptor sites on small intestinal enterocytes enabling the bacteria to colonize the small intestine [13,14]. ETEC strains equipped with F4 and F18 adhesive fimbriae show a high virulence and are the most common serotypes isolated in animals affected by PWD [15]. As an alternative to antibiotics, new compounds are urgently needed to control enteric diseases and PWD in pig livestock [14]. Tannins are suitable due to their valuable extra-nutritional properties. However, interactions with bacterial toxins seem to be specific, as only a few tannins are able to reduce ETEC diarrhea [16]. 

In the present study, we evaluated the *in vitro* antioxidant and *E. coli* F4+ and F18+ growth inhibitory activities of Qu, Ch, and their combinations in two experimental conditions: (i) After solubilization in water, to demonstrate the direct effect of bioactive compounds responsible for bacterial growth inhibition and antioxidant activities; (ii) after *in vitro* gastro-intestinal digestion to evaluate the bio-accessibility of bioactive molecules responsible for such activities. We also tested the ability of the Qu–Ch mixture to counteract H_2_O_2_ and DSS-induced stress in IPEC-J2 as a cell model of the intestinal swine epithelium.

## 2. Materials and Methods 

### 2.1. Chemical Analysis of Chestnut and Quebracho Tannin and Sample Preparation

Ch and Qu extracts tested in the present study were obtained by hot water solubilization and contain 75 g of tannin/100 g of dry matter (Silvateam S.p.A, San Michele Mondovi, Italy). 

They were tested in terms of their chemical composition (AOAC, 2005; EU regulation 152/2009) and the data are included in Appendix A.

The chemical composition of the Ch and Qu was analyzed in the laboratory for oven-dried samples (65 °C) to determine the moisture, and then ground through a 1-mm screen. Ash, crude protein (CP), neutral detergent fiber (NDF), and ether extract (EE) were determined following the methods of the Association of Official Analytical Chemists (AOAC, 2005) (Table 1) [17]. 

Ch (Ch100%), Qu (Qu100%), and three different mixtures of Ch and Qu tannins (Qu75%–Ch25%, Qu50%–Ch50% and Qu25%–Ch75%) were dissolved in hot water (100 mg/mL), neutralized (pH 7), and filter sterilized (0.22 µm filter, Millipore).

In both tannin extracts, the crude protein content was below 1.5% on a DM basis, and the fat and NDF concentrations were negligible.

Tannin mixtures were obtained by mixing the Ch100% and Qu100% powders to give the following Qu:Ch ratios: 

1:1 (Qu50%–Ch50%);

1:3 (Qu25%–Ch75%); 

3:1 (Qu75%–Ch25%).

### 2.2. Total Antioxidant Capacity—ABTS Assay

Antioxidant capacity (AOX) was determined in Ch100%, Qu100%, Qu75%–Ch25%, Qu50%–Ch50% and Qu25%–Ch75% samples (100 mg/mL) following Re et al. (1999) [18] with modifications. Trolox stock solution (2.5 mM in distilled water) was used to produce the standard curve. A solution of 2,2′-azinobis (3-ethylbenzothiazoline 6-sulfonic acid) (ABTS) (7 mM) was prepared with potassium persulfate (140 mM) in distilled water and left to react in the dark for 12–16 h to produce the ABTS•+solution. For the study of AOX capacity, the ABTS•+solution was diluted with phosphate phosphate-buffered saline, pH 7.4, (PBS) to obtain an absorbance of 0.70 (±0.02) at 734 nm and equilibrated at 30 °C. A volume of 20 µL of the sample or Trolox standard was mixed with 2 mL of ABTS•+ working solution and incubated in the dark for 6 min at room temperature before measuring absorbance at 734 nm on a spectrophotometer (Synergy HTX, Biotek). Solvent blanks were included in each assay. The percentage inhibition of absorbance at 734 nm was calculated and plotted as a function of the concentration of Trolox standard curve. AOX results were expressed as µmol Trolox equivalents (TE)/g extract.

### 2.3. E. coli Growth Inhibitory Activity 

The *E. coli* F4+ and F18+ growth inhibitory activity of Ch100%, Qu100% and Qu75%–Ch25%, Qu50%–Ch50% and Qu25%–Ch75% was evaluated for *in vitro* cultures of *E. coli* F4+ and F18+.

Two ETEC strains, harboring F4 (F4+) and F18 (F18+) adhesive fimbriae respectively, were obtained from IZSLER (Brescia, Italy). The bacteria were grown at 37 °C with shaking (150 rpm) in LB broth for 12 h prior to being used as inoculants for all experiments. F4+ and F18+ were characterized by PCR in terms of their virulence factors F4 and F18 (see the Appendix A).

Overnight-grown *E. coli* F4+ and F18+ were inoculated in tubes containing 15 mL of LB medium supplemented with 0, 200, 400, 600, 800, and 1200 μg/mL of each tannin. Prior to inoculation, the bacterial cultures were adjusted to identical densities by spectrophotometry (600 nm) across the two strains. All tubes were incubated aerobically with shaking (150 rpm) at 37 °C.

The bacterial growth was determined via measurement of the optical density of each culture at 600 nm (OD600) at 60-min intervals in a spectrophotometer (UV/VIS Lamba 365, PerkinElmer, Waltham, MA, USA). Bacterial-free tubes with equivalent concentrations of tannins were used as blanks to subtract the background turbidity caused by tannin-protein interactions [19].

All data obtained from the optical density evaluation were converted to log-transformed based cell count (CFU/mL) by a calibration curve, obtained by monitoring the *E. coli* F4+ and F18+ growth over time, in the same experimental conditions, using the classic plate counting method (data not shown) [20]. 

### 2.4. Determination of Minimal Inhibitory Concentration (MICs) and Minimal Bactericidal Concentration (MBC)

Minimum inhibitory concentrations were determined in 96-well microplates by preparing a gradient of tannin solutions (ranging from 10 mg/mL to 0 mg/mL). Briefly, 100 µL of the tannin solutions, 100 µL of LB broth and 10 µL of an *E. coli* culture (approximately 10^6^ CFU/mL) were inoculated in each well of the plate, except for the blank wells, and incubated at 37 °C for 18 h. Bacterial growth was determined by the change in absorbance after reading the microplates at 600 nm in a spectrophotometer reader (BioRAD). The MIC was defined as the lowest tannin concentration that did not produce turbidity by comparison with tannin-free control (0 mg/mL) [21]. The experiment was repeated three times and the results were expressed as average values.

*In vitro* bactericidal analyses were conducted with 0, 6, 7, 8, 9, and 10 mg/mL of Ch100%, Qu100%, Qu75%–Ch25%, Qu50%–Ch50%, and Qu25%–Ch75% at 24 h incubation in LB medium. Samples taken from all cultures were serially diluted (10-fold increments) in a sterile physiological solution. Dilutions were plated on LB agar and incubated overnight at 37 °C. Colonies grown on agar plates were directly counted after 24 h of incubation. The percentage bactericidal effect was calculated from the control vs. mg of tannins per mL. The lowest tannin concentration that did not yield any colony growth after 24-h incubation was designed as the minimum bactericidal concentration (MBC) [22].

### 2.5. Chestnut and Quebracho Tannins In Vitro Digestion and Calculation of Digestibility 

Based on antimicrobial and antioxidant results obtained in tannin water extracts, we investigated the *E. coli* growth inhibitory and antioxidant activities of Ch100%, Qu100%, and Qu50%–Ch50% after *in vitro* digestion. The digestion was performed according to the method set up and validated by Minekus et al. [23], and further adapted by our group [24,25].

Briefly, 20 g of each tannin powder (Ch100%, Qu100%, and Qu50%–Ch50%) was mixed with 150 mL of distilled H_2_O and kept on an orbital shaker at 150 rpm for 5 min. The digestion procedure involved three phases. For the oral phase, 6.66 mg α-amylase in 2.1 mL of 1 mM CaCl_2_, pH 7 was added to the samples which were then incubated for 30 min at 37 °C on a shaker. For the gastric phase, the pH was decreased to 2 with 6 M HCl and 0.9 g of pepsin was added in 8.3 mL of 0.1 M HCl. The samples were then incubated for 120 min at 37 °C on a shaker. For the small intestinal phase, the pH was increased to 7 with 6 M NaOH and 0.2 mg pancreatin and 1.2 g bile in NaHCO_3_ 0.5 M were added to the samples before carrying out the final incubation of 180 min at 37 °C on a shaker. 

A blank sample (enzymes of the digestion alone), along with a positive and negative control were included as reference samples in all the digestions performed (*n* = 3).

At the end of digestion, the total digesta obtained was transferred to a 3-kDa cutoff membrane (Vivaspin 20, Sartorius, Göttingen, Germany). Each filter was previously activated with 0.1% BSA solution. Samples were centrifuged for 20 min at 3500× *g* (5 °C). Aliquots from the filtrate were sampled and snap-frozen in liquid nitrogen to stop enzyme activity, before storing at −80 °C for further experiments.

The undigested fraction was collected and used to calculate *in vitro* digestibility as detailed by Castrica et al. [24].

### 2.6. Antioxidant and E. coli Growth Inhibitory Activities of In Vitro Digested Tannins

Antioxidant and *E. coli* F4+ and F18+ growth inhibitory activity activities of physiological extracts of Ch100%, Qu100%, and Qu50%–Ch50% were performed as described above.

### 2.7. Effects of In Vitro Digested Chestnut and Quebracho Tannins on Intestinal IPEC-J2 Cell Viability

IPEC-J2 cells are intestinal porcine enterocytes isolated from the jejunum of a neonatal unsuckled piglet (ACC 701, DSMZ, Braunschweig, Germany). The IPEC-J2 cell line is unique as it is derived from the small intestine and is not transformed nor tumorigenic in nature [26]. The IPEC-J2 cells were cultured in DMEM/F-12 mix (Dulbecco’s Modified Eagle Medium, Ham’s F-12 mixture) supplemented with HEPES, fetal bovine serum (FBS), penicillin/streptomycin and cultivated in a humid chamber at 37 °C with 5% CO_2_. All experiments were performed using IPEC-J2 cells within six cell passages (passages 16 to 22) to ensure reproducibility.

IPEC-J2 cells were seeded at a density of 1.5–2 × 10^5^ cells/mL in 96-well plates and cultured for 24 h. In addition, dose-response curves (cell viability) of *in vitro* digested Ch, Qu, and Qu50%–Ch50% were performed on IPEC-J2 cells based on preliminary experiments on bacteria (0–1200 µg/mL) obtained in previous experiments. Cell viability was determined after three hours of tannin treatment by a colorimetric proliferation assay (MTT test) in accordance with the manufacturer’s instructions.

In a second set of experiments, IPEC-J2 cells at sub-confluence were pre-treated with Ch, Qu, and Qu50%–Ch50% for three hours, and further challenged with H_2_O_2_ or with DSS to induce chemical stress in the cell culture. Hydrogen peroxide was applied at a concentration of 0.5 mM for 1-h incubation. DSS at a concentration of 2% was applied for 24 h. Time and doses of H_2_O_2_ incubation were based on preliminary data and on the literature [27]. Time and doses of DSS incubation were based on our preliminary study, where IC50 was calculated for IPEC-J2 cells [28].

### 2.8. Statistical Analysis

Statistical analysis was performed using GraphPad-Prism 8. *E. coli* growth data (OD600) were log10 transformed prior to statistical analysis. *E. coli* growth data were subjected to analysis of variance using the MIXED procedure. The model included the fixed effect of treatments (Ch100%, Qu100%, Qu75%–Ch25%, Qu50%–Ch50%, and Qu25%–Ch75%), time and time x treatment. One-way ANOVA was used to analyze antioxidant and cell viability data. The differences between means were compared using Tukey’s test and considered statistically significant at *p* < 0.05. Data are presented as least square means ± SEM.

## 3. Results

### 3.1. Antioxidant and E. coli F4+ and F18+ Growth Inhibitory Activity of Chestnut and Quebracho Tannin Water Extracts

#### 3.1.1. Total Antioxidant Capacity—ABTS Assay

Chestnut and Qu tannin extracts showed an antioxidant (AOX) capacity. Among the tested samples, Qu25%–Ch75% showed the highest AOX capacity (6860 ± 121.9 µmol TE/g tannin powder). Ch100% exhibited higher AOX compared with Qu100% (5243.33 ± 113.1 vs. 3164.81 ± 166.2 µmol TE/g), which showed the lowest AOX capacity compared with all the other samples tested (Figure 1). The AOX capacity of Ch100% and Qu50%–Ch50% was comparable (*p* > 0.05). Trolox at a concentration of 2000 µM was included as an internal control and showed an AOX capacity of 1.828 µmol TE/g.

#### 3.1.2. *E. coli* Growth Inhibitory Activity

The growth of *E. coli* F4+ and F18+ strains was tested in the absence or presence of different concentrations (0–1200 µg/mL) of Ch and Qu tannin extracts (see Appendix A). In general, a dose-dependent effect was observed at each time point for both Ch and Qu treatments, for which, the maximum growth inhibition was observed at a concentration of 1200 µg/mL. Based on this result and on previous unpublished data on the synergistic effect of tannin extracts, we therefore compared the effects of Qu, Ch and of different combinations of Ch and Qu (Qu75%–Ch25%, Qu50%–Ch50%, and Qu25%–Ch75%) at a concentration of 1200 µg/mL (Figure 2) in *E. coli* F4+ and F18+. Generally, in both F4+ and F18+ the combined use of Ch and Qu showed a synergistic activity in the inhibition of F4+ and F18+ growth. Overall, the combinations with the highest Ch concentration (Qu50%–Ch50% and Qu25%–Ch75%) were the most effective (Figure 2).

#### 3.1.3. Determination of Minimal Inhibitory Concentrations (MICs) and Minimal Bactericidal Concentration (MBC)

The minimal inhibitory concentration (MIC) and minimal bactericidal concentration (MBC) of Qu and Ch tannins and of their combinations were calculated in *E. coli* F4+ and F18+ (Table 2). Data showed a similar bactericidal activity of both Ch and Qu, in particular in tannin combinations. The bactericidal effect of 6 mg/mL of Qu and Ch tannins was partial (55–62% for F4+ and 50–60% for F18+) after 24 h of incubation. In contrast, there was a complete bactericidal effect when Ch and Qu were evaluated, after 24 h of incubation, at a concentration of 9 mg/mL for *E. coli* F4+ and 8 mg/mL for *E. coli* F18+ (Table 2).

### 3.2. Antioxidant and E. coli Growth Inhibitory Activities of In Vitro Digested Chestnut and Quebracho Tannin

Qu100%, Ch100%, and Qu50%–Ch50% were *in vitro* digested and subsequently tested for their antioxidant, *E. coli* growth and cytomodulatory activity in IPEC-J2 cells. The combination Qu50%–Ch50% was also tested on IPEC-J2 cells subjected to stress chemically induced by H_2_O_2_ and DSS. The *in vitro* digestion enabled the digestibility values to be calculated for Ch and Qu, which corresponded to 66.16% (of DM) and 71.93% (of DM) for Ch and Qu, respectively.

#### 3.2.1. Total Antioxidant Capacity—ABTS Assay

Physiological extracts of Ch and Qu tannins showed 2433.33 ± 114. 15 and 1944. 81 ± 151.95 µmol TE/g. Physiological extracts of Qu50%–Ch50% showed an AOX of 2434.76 ± 211.80 µmol TE/g (Figure 3). 

#### 3.2.2. *E. coli* Growth Inhibitory Activity

The activity of Ch, Qu, and Qu50%–Ch50% on *E. coli* growth was further tested after *in vitro* digestion (Figure 4).

The *E. coli* growth inhibitory activity of Qu on was significantly higher compared to all other treatments from T1 to T6 (*p* < 0.05) for F4+, but was significantly higher compared to all other treatments from T2 to T6 (*p* < 0.05) for F18+. Qu50%–Ch50% activity was significantly higher from T2 to T6 for F4+ compared with the control, and was significantly higher (*p* < 0.05) from T1 to T4 for F18+. Ch, however, did not show a significant *E. coli* growth inhibitory activity after *in vitro* digestion.

#### 3.2.3. Effect of *In Vitro* Digested Chestnut and Quebracho Tannin Extracts on IPEC-J2 Cells Chemically Challenged with H_2_O_2_ and DSS

The *in vitro* digested Qu, Ch, and Qu50%–Ch50% were also tested for swine intestinal epithelial IPEC-J2 to determine whether tannins also affect the viability of the cells. Dose-response curves with several concentrations of *in vitro* digested Qu, Ch, and Qu50%–Ch50% tannins were tested on IPEC-J2 cells and the viability was assessed after three hours of incubation (Figure 5). 

The results showed that at the highest concentrations of *in vitro* digested Ch, Qu, and Qu50%–Ch50% tested (1200–600 µg/mL), IPEC-J2 cell viability was significantly reduced compared with 0 µg/mL; at a concentration of 400 µg/mL, Qu50%–Ch50% significantly increased IPEC-J2 cell viability, Ch and Que reduced cell viability (*p* < 0.05). At the lowest concentrations tested (200–50 µg/mL), IPEC-J2 cell viability was unaltered or increased by Ch and Qu treatment and was significantly increased by Qu50%–Ch50% treatment (Figure 5). Qu50%–Ch50% was the most effective treatment and stimulated IPEC-J2 cell viability after three hours of incubation. Based on the trophic effect observed in IPEC-J2 cells treated with *in vitro* digested Qu50%–Ch50%, this combination was chosen for the cell-challenging experiments. 

We also tested the trophic effect of Qu50%–Ch50% on IPEC-J2 cells previously stressed with H_2_O_2_ or DSS. In particular, IPEC-J2 cells were pre-treated for three hours with *in vitro* digested Qu50%–Ch50% at different concentrations (50–1200 µg/mL) and further challenged with H_2_O_2_ for 1 h or with DSS for 24 h, to simulate *in vitro* conditions of oxidative and inflammatory stress at the level of the intestinal cell epithelium.

In the DSS-challenged IPEC-J2 cells (Figure 6a), the lowest concentrations of Qu50%–Ch50% (50–400 µg/mL) significantly counteracted DSS-induced stress by increasing cell viability. However, at concentrations from 600–1200 µg/mL Qu50%–Ch50% did not counteract DSS-induced stress.

In the H_2_O_2_ -challenged IPEC-J2 cells (Figure 6b), the 3-h pre-treatment with Qu50%–Ch50% mitigated the oxidative stress experimentally induced by increasing cell viability. In particular, Qu50%–Ch50% at the highest concentrations tested (1200–200 µg/mL) significantly counteracted the stress induced by 0.5 mM of H_2_O_2_. 

## 4. Discussion

The principal objectives of this study were to assess the AOX capacity and *E. coli* growth inhibitory activity of Qu, Ch, and their mixtures after solubilization in water and after *in vitro* digestion. We also aimed to determine whether tannin digests have a trophic effect on swine intestinal epithelial IPEC-J2 cell viability. 

The AOX capacity results from our study revealed that Ch and Qu25%–Ch75% showed the highest AOX capacity compared with Qu and with all the other tannin mixtures. In the combined samples, the presence of Ch dose-dependently increased the AOX capacity and the combination of high doses of Ch (75%) with low doses of Qu (25%) showed the highest AOX effect. Comparing our data with those reported by Pèrez-Burillo et al. [29,30], our samples showed an over 30-fold higher antioxidant capacity than the conventional food and feed ingredients analyzed in their study. The high reducing capacity of Ch and Qu could be attributed to the high concentration of phenolic compounds in tannin extracts. On the other hand, *in vitro* digested Ch and Qu50%–Ch50% showed a higher antioxidant activity compared to Qu. Our data are in line with those of Molino et al. [31] who reported a higher antioxidant activity for Ch (8.16 mmol Trolox/g) compared with Qu (6.70 mmol Trolox/g) after *in vitro* digestion. In general, compared to our results, they reported higher values of antioxidant capacity in their samples, which may be due to the different methodology used for antioxidant evaluation (GAR method) and the different *in vitro* digestion protocol used in their study. 

Our *E. coli* growth inhibitory activity results demonstrated that Ch and Qu tannins, at specific concentrations and time, inhibit the growth of *E. coli* F4+ and F18+ *in vitro*. The Ch rapidly (from T1) became effective, while Qu seemed to exhibit a stable growth inhibitory activity only after three hours of incubation with *E. coli*. However, Qu *E. coli* growth inhibitory activity was maintained until the end of our analysis (6 h). 

These results suggest that the rapid effect of Ch observed was associated with the more prolonged effect of Qu. Generally, under our culture conditions, the combined use of Ch and Qu had a synergistic activity in the inhibition of F4+ and F18+ growth. Overall, the combinations with the highest Ch concentration (Qu50%–Ch50% and Qu25%–Ch75%) were the most effective. However, we selected the combination Qu50%–Ch50% for further analysis as it represents a balance of Ch and Qu in which the faster activity of Ch and the more prolonged activity of Qu over time are combined. 

In addition, the *E. coli* growth inhibitory activity of Qu and of Qu50%–Ch50% against F4+ and F18+ was maintained after *in vitro* digestion, thus highlighting the possible bio-accessibility of the antimicrobial compounds in these samples. However, the *E. coli* growth inhibitory activity of Ch was not maintained after digestion, which may be due to a lower bio-accessibility or to the excessive degradation of antimicrobial and antioxidant molecules in our experimental conditions. 

The *E. coli* growth inhibitory activity of Ch and Qu has been evaluated in several studies. Min et al. [19] demonstrated that chestnut and mimosa tannins have growth-inhibitory and bactericidal effects *in vitro* against *E. coli* O157:H7, and chestnut tannins showed a higher bactericidal activity. Elizondo et al. [32] found that the antibacterial and antioxidant activities of Ch added to Qu tannin were higher than pure Qu but lower than Ch tannin alone. They concluded that although Ch tannin is more potent than Qu tannin, the Qu activity may remain longer in the gastrointestinal tract because of their rich condensed tannin composition. This latter point was corroborated by our results. 

In fact, tannins can inhibit the growth of some pathogenic bacterial species (e.g., *E. coli*) without affecting the physiological growth and proliferation of probiotic lactic acid bacteria, which have a positive effect at the intestinal level [33]. This selective effect may be an advantage in the use of tannins in feed. The *E. coli* growth inhibition and AOX capacity of tannins may be due to the high levels of phenols in the extracts. In our experimental conditions, these functional activities were often maintained after digestion. However, these data need to be confirmed in further *in vitro* digestion tests and after total phenolic compound analysis in the digesta. 

We also tested *in vitro* digested Qu, Ch and Qu50%–Ch50% on swine IPEC-J2 cell viability to determine whether *E. coli* growth inhibitory concentrations also affect the viability of the intestinal cells. Qu50%–Ch50% was the most effective in stimulating cell viability when administered at low concentrations. It showed a trophic effect on IPEC-J2 cells, and was therefore, used for the cell-challenging experiments. 

We found that a pre-treatment of three hours with Qu50%–Ch50% mitigated the mild oxidative and inflammatory stress experimentally induced in IPEC-J2 cells. This thus highlights the potential of tannins in preventing oxidative and inflammatory conditions at the intestinal level. However, the possibility that the lowest concentrations of tannins were effective in the DSS challenge, while only the highest concentrations were effective in counteracting oxidative challenge (H_2_O_2_) still needs to be investigated. We suggest that the combined treatment of IPEC-J2 cells with tannins and H_2_O_2_ versus tannins and DSS may show a different behavior and synergism. A limited number of studies have investigated the ability of tannins to affect intestinal cell proliferation. Brus et al. [34] reported that low doses of gallic acid increased the proliferation of IPEC-J2 cells, thus highlighting the possible role of gallic acid in the recovery of small intestinal epithelium in swine. The same group reported the ability of several commercial products containing tannins in stimulating the proliferation of IPEC-J2 and Caco-2 cells, although at lower doses compared to our study. They also reported [35] that a water-soluble form of Ch tannin exerted beneficial effects on the small intestinal epithelial cells of chickens by stimulating the proliferation of enterocytes and increasing the antioxidant potential, with no adverse effects on cellular metabolism. Cell models of the intestine are useful as *in vitro* tools for assessing feed and food ingredients [36,37,38,39] as they represent a simplified version of the *in vivo* intestinal environment. For the safe use of tannin-based additives in feed and food, *in vitro* animal cell models can be used as a cheap and practical alternative to animal experiments to estimate the optimal dosage for further practical use. 

We believe that our results corroborate the potential beneficial use of Ch and Qu, in particular in combination, in the animal diet as antimicrobial and antioxidant agents. In addition, they have a trophic effect on intestinal epithelial cells. 

## 5. Conclusions

Our data demonstrate the ability of quebracho and chestnut tannin to exert antioxidant activity and *E. coli* growth inhibitory activity against ETEC F4+ and F18+, together with cyto-protective activity on swine intestinal epithelial cells at specific doses.

After *in vitro* digestion, chestnut and quebracho showed an antioxidant capacity and above all quebracho maintained its *E. coli* growth inhibitory activity. Our data clearly demonstrate that although chestnut and quebracho had a higher *E. coli* growth inhibitory effect when administered at a high dosage, they have a trophic effect on the intestinal cell epithelium also when used at lower dosages. Based on our findings from IPEC-J2 culture, we conclude that a balanced combination of tannins (1:1) at specific dosages may produce a protective and stimulating effect on cell proliferation rather than a cytotoxic effect. 

Besides suggesting the combined use of chestnut and quebracho as a strategy to maximize the inner functional effects of such compounds, our findings also indicate that the actual dosage of tannins may be key in determining their effect on bacteria and cells. Our data provide the basis for further *in vivo* studies aimed at optimizing the use of tannins as functional feed and food ingredients.

## Figures and Tables

**Figure 1 animals-10-00195-f001:**
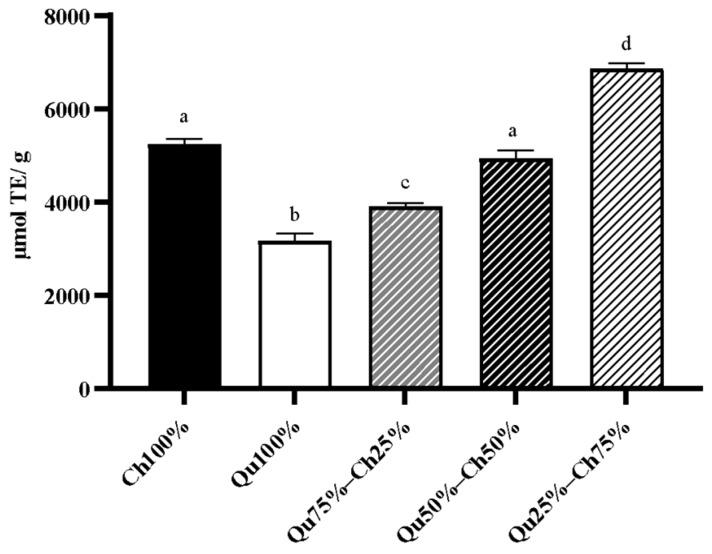
Antioxidant capacity of Ch100%, Qu100%, Qu75%–Ch25%, Qu50%–Ch50%, and Qu25%–Ch75% (100 mg/mL). Data are presented as lsmeans ± SEM (*n* = 3). The different superscript letters indicate a significant difference at *p* < 0.05 (one-way ANOVA). 2000 µM Trolox was included as an internal control.

**Figure 2 animals-10-00195-f002:**
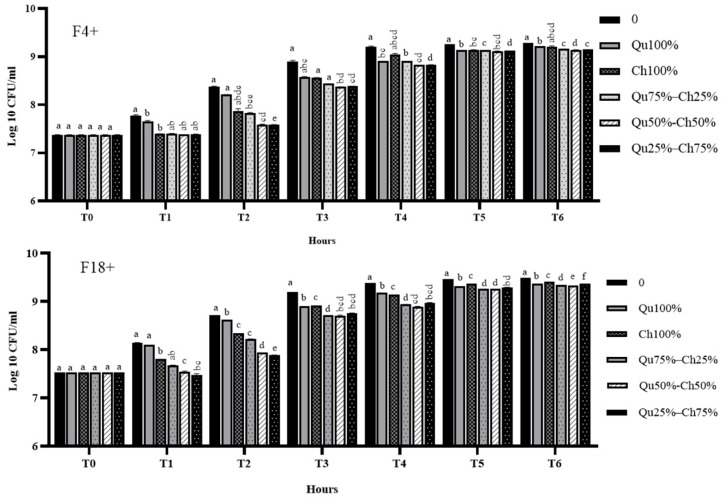
Effects of 1200 µg/mL of Qu (Qu100%), Ch (Ch100%), Qu75%–Ch25%, Qu50%–Ch50%, and Qu25%–Ch75% on *E. coli* F4+ and F18+ growth over time (T). Data are expressed as log10 CFU/mL lsmeans ± S.E.M. (*n* = 3, mixed ANOVA). Different superscript letters indicate significant differences at *p* < 0.05 among different concentrations within the same time point.

**Figure 3 animals-10-00195-f003:**
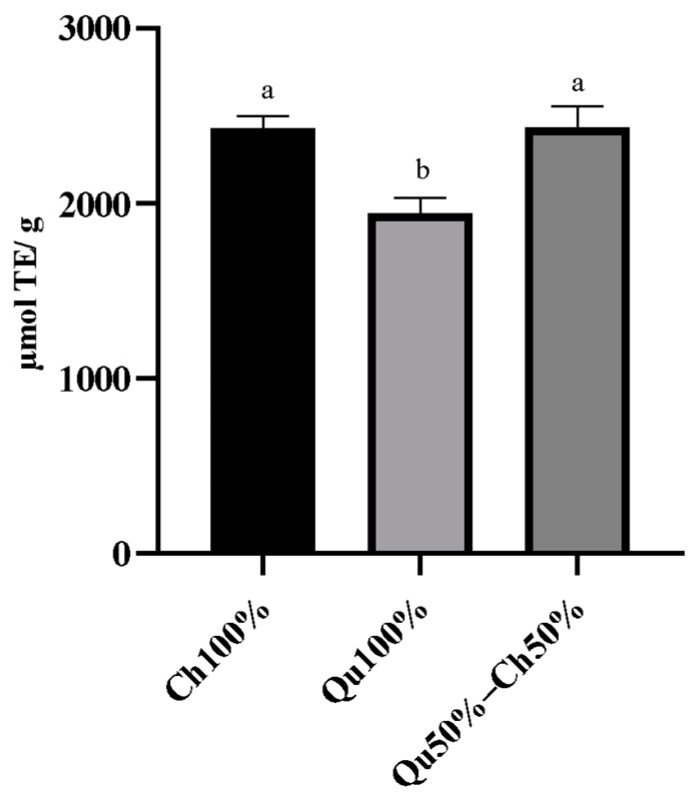
Antioxidant capacity of *in vitro* digested Ch100%, Qu100% and Qu50%–Ch50%. Data are presented as lsmeans ± SEM (*n* = 3). The different superscript letters indicate a significant difference at *p* < 0.05 (one-way ANOVA). 2000 µM Trolox was included as an internal control.

**Figure 4 animals-10-00195-f004:**
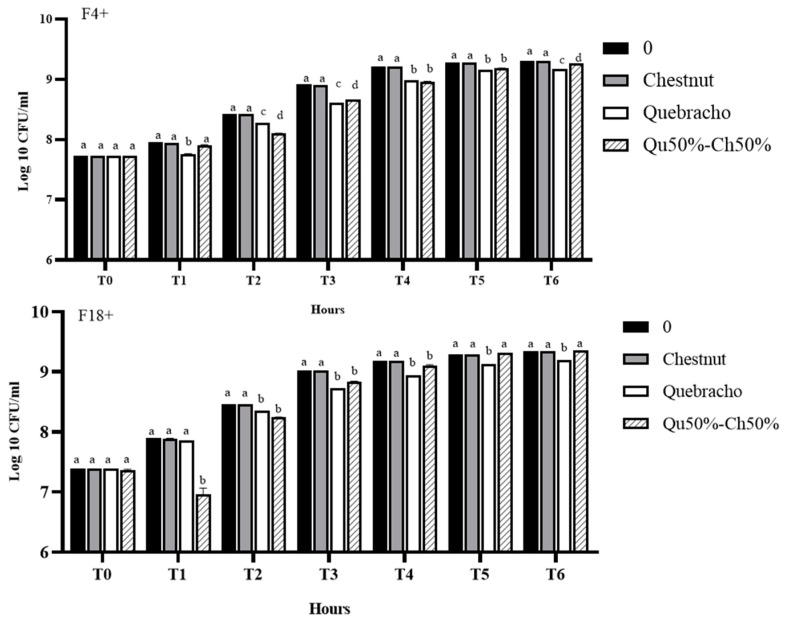
Effects of 1200 µg/mL of Ch 100%, Qu 100%, and Qu50%–Ch50% after *in vitro* digestion on *E. coli* F4+ and F18+ growth over time. Data are expressed as log10 CFU/mL lsmeans ± S.E.M. (*n* = 3, mixed ANOVA). Different superscript letters indicate significant differences at *p* < 0.05 among different concentrations within the same timepoint.

**Figure 5 animals-10-00195-f005:**
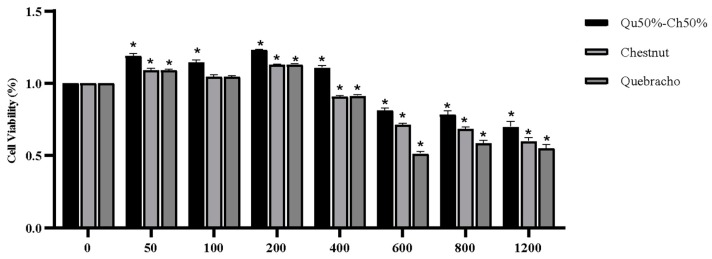
Effect of different concentrations (0–1200 µg/mL) of *in vitro* digested Qu50%–Ch50% on IPEC-J2 cell metabolic activity (expressed as cell viability, MTT assay). Data are expressed as lsmeans ± SEM (*n* = 3, one-way ANOVA). * indicates significant differences at *p* < 0.05 compared to respective control wells (0 µg/mL).

**Figure 6 animals-10-00195-f006:**
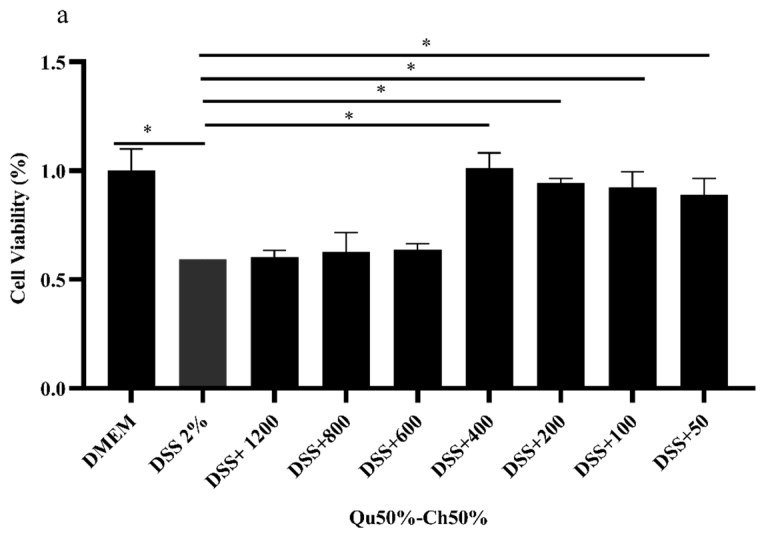
Effect of 3-h pre-treatment with different concentrations (50–1200 µg/mL) of *in vitro* digested Qu50%–Ch50% on IPEC-J2 cells further stressed with 2% DSS for 24 h (**a**) or 0.5 mM of hydrogen peroxide (H_2_O_2_) (**b**) on the metabolic activity (expressed as cell viability, MTT assay). Data are expressed as lsmeans ± SEM (*n* = 3, one-way ANOVA). * Denote significant differences (*p* < 0.05).

**Table 1 animals-10-00195-t001:** Chemical composition of Qu and Ch tannins.

% on Dry Matter	Moisture EU Regulation 152/2009	Ash AOAC 942.05 (2005)	Neutral Detergent Fiber (NDF) AOAC 2002.04 (2005)	Crude Protein AOAC 2001.11 (2005)	Ether Extract AOAC 2001.11 (2005)
Qu	4.82 ± 0.04	2.01 ± 0.15	<0.5	1.40 ± 0.02	<0.5
Ch	5.30 ± 0.04	0.99 ± 0.05	<0.5	0.90 ± 0.1	<0.5

**Table 2 animals-10-00195-t002:** The minimal inhibitory concentration (MIC) and minimal bactericidal concentration (MBC) of Qu and chestnut tannins on *E. coli* F4+ and F18+.

Bacteria	MIC (mg/mL)					MBC (mg/mL)				
	Qu100%	Ch100%	Qu75%–Ch25%	Qu50%–Ch50%	Qu25%–Ch75%	Qu100%	Ch100%	Qu75%–Ch25%	Qu50%–Ch50%	Qu25%–Ch75%
*E. coli* F4+	6	7	6	6	6	9	9	8	8	8
*E. coli* F18+	6	7	6	6	6	8	8	8	8	8

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
