# Peer review of "In Vitro Digestion of Chestnut and Quebracho Tannin Extracts: Antimicrobial Effect, Antioxidant Capacity and Cytomodulatory Activity in Swine Intestinal IPEC-J2 Cells"

_animals, 2020, doi:10.3390/ani10020195_

Round 1

Reviewer 1 Report

The principal objectives of this study were to assess the AOX capacity and antimicrobial activity of Qu, Ch and their mixtures before and after in vitro digestion, however, there were no results on the comparison before and after in vitro digestion.

Author Response

Reviewer 1

The principal objectives of this study were to assess the AOX capacity and antimicrobial activity of Qu, Ch and their mixtures before and after in vitro digestion, however, there were no results on the comparison before and after in vitro digestion.

Many thanks for your comment.

the actual objective of the manuscript is to investigate the antioxidant and E. coli F4+ and F18+ growth-inhibitory activity of Qu, Ch and their combinations in two experimental conditions: after water extraction, to demonstrate the direct bioaccessibility of bioactive compounds responsible to antimicrobial and antioxidant activities; after simulated gastro-intestinal digestion in vitro to simulate the digestion and the physiological liberation of effective bioactive molecules. The authors do not want to compare before and after digestion as the sample condition and characteristics differ drastically due to different methods of extraction.

However, according to reviewer suggestion we have adapted the aim of the study to not mislead the reader. The aim was changed into: “In the present study, we evaluated the in vitro antioxidant and E. coli F4+ and F18+ growth inhibitory activities of Qu, Ch and their combinations in two experimental conditions: Ii) after solubilization in water, to demonstrate the direct effect of bioactive compounds responsible for bacterial growth inhibition and antioxidant activities; ii) after in vitro gastro-intestinal digestion to simulate the bio-accessibility of bioactive molecules responsible for such activities. We also tested the ability of the Qu-Ch mixture to counteract oxidative and inflammatory stress in IPEC-J2 as a cell model of the intestinal swine epithelium"

The authors have also changed the discussion to enphasize the differences among activity after water solubilization and in vitro digestion, according to Reviewer suggestion.

Reviewer 2 Report

Manuscript: In vitro digestion of chestnut and quebracho tannin extracts: antimicrobial effect, antioxidant capacity and cytomodulatory activity in swine intestinal IPEC-J2 cells
In the current manuscript, the authors investigated the antioxidant and antimicrobial activity of Qu and Ch in E. coli and intestinal epithelial IPEC-J2 cells. They found that undigested Qu and Ch alone and in combination showed antioxidant capacity. They also reported that digested Qu50%-Ch50% counteracted the hydrogen peroxide and DSS experimentally-induced stress in the intestinal IPEC-J2 cells. Overall, this work provided the preliminary data for further in vivo studies that is to optimize the use of tannins as functional feed and food ingredients in animal feeding. Some concerns should be addressed before publication.
Major concerns: 1 ABTS assay in the 3.2.1 section did not show Figure.
2 All Figures, except Figure1 and Figure5, have to be improved in the high resolution and labeled clearly. And Figure Legends have to put explanations for each label. Besides, the statistical comparison should be denoted on all of the figures and keep the consistent labels in all figures.
3 What is the big difference between undigested and digested Qu/Ch in elements? The authors should discuss more the difference.
Minor concerns: a) Careful editing is needed. And the full version has to rearrange, especially methods and results sections. The whole article has to show each section and Figure before publication clearly.

Author Response

Reviewer 2

Manuscript: In vitro digestion of chestnut and quebracho tannin extracts: antimicrobial effect, antioxidant capacity and cytomodulatory activity in swine intestinal IPEC-J2 cells
In the current manuscript, the authors investigated the antioxidant and antimicrobial activity of Qu and Ch in E. coli and intestinal epithelial IPEC-J2 cells. They found that undigested Qu and Ch alone and in combination showed antioxidant capacity. They also reported that digested Qu50%-Ch50% counteracted the hydrogen peroxide and DSS experimentally-induced stress in the intestinal IPEC-J2 cells. Overall, this work provided the preliminary data for further in vivo studies that is to optimize the use of tannins as functional feed and food ingredients in animal feeding. Some concerns should be addressed before publication.

Major concerns:

1 ABTS assay in the 3.2.1 section did not show Figure.

Thanks. We have added a graph of OAX rasults.

2 All Figures, except Figure1 and Figure5, have to be improved in the high resolution and labeled clearly. And Figure Legends have to put explanations for each label. Besides, the statistical comparison should be denoted on all of the figures and keep the consistent labels in all figures.

Figure 1 was placed in supplementary material at higher resolution. All other figures were modified to increased resolution. And statistical analysis was added to each legend.

3 What is the big difference between undigested and digested Qu/Ch in elements? The authors should discuss more the difference.

Many thanks for your comment. However, the actual objective of the manuscript is to investigate the antioxidant and antimicrobial activity against E. coli F4+ and F18+ growth of Qu, Ch and their combinations in two experimental conditions: after water extraction, to demonstrate the direct bioaccessibility of bioactive compounds responsible to antimicrobial and antioxidant activities; after simulated gastro-intestinal digestion in vitro to simulate the digestion and the physiological liberation of effective bioactive molecules.. The authors do not want to compare before and after digestion as the sample condition and characteristics differ drastically. We change the discussion and aim of the study to underline the differences and discuss the data.

Minor concerns: a) Careful editing is needed. And the full version has to rearrange, especially methods and results sections. The whole article has to show each section and Figure before publication clearly.

Thanks for the suggestion, the manuscript was improved, material and methods and figures were carefully checked, adapted to manuscript content. (e.g. Thanks. A separate paragraph was added to the results section with MIC and mBC data)

Reviewer 3 Report

This manuscript deals with the evaluation of the in vitro antioxidant, antimicrobial and cytomodulatory effects of two different undigested or digested natural extracts rich in tannins. The overall result sounds interesting and capable to complement data obtained by other in vivo studies in the field, however the experimental design should be improved. Moreover, the manuscript should be revised for its English, together with many paragraphs and statements, that would need a major revision.

Introduction

Lines 60-61 – Attention should be payed to the absorption of these compounds and when/where does it occur all across the digestive tract, in order to understand the importance of the in vitro digestion that is performed in this experiment.

Line 66 – The word “post” in post-weaning diarrhea appears two times instead of one.

Line 74 – Reference 16 links to an article in bibliography that seems not to deal with information reported in the sentence it is linked to.

Line 76 – Adequate references should be added to the statement.

Materials and Methods

Line 84 – Besides chemical composition, since this manuscript widely refers to tannins, it would be interesting to provide data regarding the composition in tannins of these extracts and their abundance. This could also help in understanding the different behaviours exerted by the two extracts, both before and after digestion.

Line 114 – The method of expression the AOX results should be slightly clarified.

Line 133 – It is not clear if provided informations are referred to the measurement of bacterial growth with tannins or to the calibration curve, which should be explained more extensively. More generally, this entire paragraph should be re-written a little more clearly.

Line 136 – The entire paragraph regarding MIC tests should be completely re-written and better explained. The scheme for the preparation of stock solution dilutions and the plates is not clear and all the assessed concentrations are not reported. Moreover, it is not clear why 200µL of stock solutions were added to the first wells of each column (already containing 100µL BHI?), a volume of stock solution that appears to be too much high if compared to the BHI volume. In addition, the incubation time of the plates before absorbance reading is not reported. Finally, giving the method used, MIC should be considered as the lowest concentration of a substance that, in this case, gives null absorbance.

Line 153 – The purpose of reporting how to calculate the percentage of bactericidal effect is not clear, because it is not shown in results.

Line 161 – The starting material of in vitro digestion should be clarified: is it the starting product or the one already dissolved in water?

Line 182 – Adequate references should be added to sentences introducing IPEC-J2 cells.

Lines 186-188 – Cultivation conditions should only be stated once.

Line 191 – “dose dose-response” should be corrected. Moreover, in this paragraph, assessed concentrations are not reported. Moreover, treatment conditions are not reported. Furthermore, parameters measured (vitality) and the test used to create the “dose-response curve” are not clarified.

Line 194 – “in a second the second” should be corrected. Moreover, in this second set of experiments, it is not clear which parameter is assessed and by means of which test.

Line 196 – The hydrogen peroxide concentration of 1 mM differs from the 0.5 mM reported in the results.

Results

Line 213 – Unit of measurement for the provided data should be corrected.

Figure 1 – In the Y axis, correct umol with µmol. Moreover, the Y axis legend should correspond to what is reported in the text.

Line 227 – Correct ug with µg.

Figure 2 – The quality of the image provided is too low, so it is very difficult to analyse data. Moreover, authors should find a clearer way to present these data.

Lines 229-240 – The presented differences between groups seem to be very small and limited to the same logarithm for each group. These data cannot support – at those tested concentrations – an antimicrobial activity of both extracts. Even if statistical difference is shown, an antimicrobial property should be demonstrated by a complete lack of growth of the bacterium, while these data not only demonstrate the capacity of the microorganisms to grow with the addition of tannin extracts, but also to reach, at the end of the experiment, the same log10(CFU/mL) of the controls. Finally, if the MIC is found at 6000-7000 mg/L, it should not be possible to observe an antimicrobial effect at lower doses. All these data should be completely reconsidered and re-interpreted, if not removed until further analysis.

Line 243 – The incipit “In F4+ and F18+” seems to introduce readers to comments regarding Figure 3, which are not reported.

Figure 3 – As reported for data shown in figure 2, these data do not support an antimicrobial activity of these compound at 1200 mg/L, even if some little statistically significant differences are reported. In fact, in the end, the addition of extracts and their combination brings all experimental groups to the same log10(CFU/mL). All these data should be completely reconsidered and re-interpreted, if not removed until further analysis.

Line 249 – MIC and MBC results should be reported in a new paragraph with a separate title, as happens for materials and methods explanations.

Lines 270-272 – Spacing between symbols, letters and numbers should be reviewed in this paragraph. Moreover, as happens in figure 1, all data should be presented in a graph (with statistical analysis) and appropriate comparison of AOX properties between digested and undigested extracts should be added.

Figure 4 - As reported for data shown in figure 1 and 2, these data do not support an antimicrobial activity of these digested compounds at 1200 mg/L, even if some little statistically significant differences are reported. In fact, in the end, the addition of extracts and their combination brings all experimental groups to the same log10(CFU/mL). Moreover, an antimicrobial activity should not only reduce bacterial growth rate, but totally prevent it. All these data should be completely reconsidered and re-interpreted, if not removed until further analysis.

Figure 5 – “Different superscript letter” in caption should be corrected appropriately; caption should also report n (experimental replicates). Authors should revise comments, since it is reported that the 400µg/mL dose does not alter viability, even if a statistically significant reduction is shown in the graph for some groups. Moreover, authors should also comment why the 100µg/mL dose of Che and Qu extracts do not statistically significant increase vitality, while 50 and 200µg/mL result in a significant increase. Furthermore, in the comments, a “CTR DMEM” group is mentioned, but not clearly shown in the graph. Finally, statistical comparison between groups is not clear, thus it should be improved.

Lines 305-309 – This paragraph is confusing because it explains the subsequent test in two different ways, so it is not clear if cells were previously treated and then challenged, or if the challenge occurred before the treatment.

Figure 6 – Caption should contain n (experimental replicates). While interesting effects are clearly proved with the DSS challenge, no statistically significant difference is shown between the “DMEM” and “0.5mM H2O2” groups, so it seems that the challenge could not significantly alter cell viability. For this reason, all the other results cannot be considered as a mitigation of the oxidative stress induced by H2O2. Moreover, authors should also comment why the same Qu50-Che50 levels (1200-800-600µg/mL) that improve cell viability over 100% in the H2O2 challenge, in Figure 5 exerted a significant reduction in cell viability.

Lines 315 – 321 – Authors should comment on the capacity of a vitality test to give information about a reduction of oxidative stress.

Discussion

Lines 328-329 – This statement is not correct, because Ch had a significant lower antioxidant capacity when compared to Qu25-Ch75 (Figure 1).

Line 334 – Adequate reference should be provided.

Lines 341-359 – Discussion should be revised considering that antimicrobial data of growth experiments with doses equal to or lower than 1200 mg/L do not prove an antimicrobial activity of the two extracts or mixtures.

Line 381 – The [26] reference seems to link to a wrong article.

Lines 383-385 – Reference should be provided.

Lines 391-396 – The two sentences tell the same concept, so it should be condensed in only one sentence.

Throughout the discussion, an explanation of which compounds could be responsible for the antimicrobial effect is missing, and it is not clear if the same antioxidant molecules in the extract could be considered responsible as well of the antimicrobial activity.

Conclusions

Conclusion should be properly adjusted considering that antimicrobial data of growth experiments with doses equal to or lower than 1200 mg/L do not prove an antimicrobial activity of the two extracts or mixtures.

Supplementary material

Line 419 – It is not clear if primers were chosen from sequences published in other articles. If so, adequate references should be provided for each primer.

Table S2 – Melting temperature for each primer should be provided.

Abstract and summary

Abstract and summary should be revised considering all the comments made to the article text.

Author Response

Reviewer 3

This manuscript deals with the evaluation of the in vitro antioxidant, antimicrobial and cytomodulatory effects of two different undigested or digested natural extracts rich in tannins. The overall result sounds interesting and capable to complement data obtained by other in vivo studies in the field; however the experimental design should be improved. Moreover, the manuscript should be revised for its English, together with many paragraphs and statements that would need a major revision.

Introduction

Lines 60-61 – Attention should be payed to the absorption of these compounds and when/where does it occur all across the digestive tract, in order to understand the importance of the in vitro digestion that is performed in this experiment.

Thanks for the comment. The absorption of tannins occurs mainly in the small intestine at the duodenal and jejunal level. Therefore, we chose to test the digested samples in IPEC-J2 cells which a model of jejunal epithelium in swine and the first target of digesta. According to reviewer suggestion, we added a statement in the introduction to better clarify the concept and to valorise the use of this cell model. The English was also edited and proofread by E4AC on 28th December 2019 (English for academics sas of Adrian John WALLWORK.

Line 66 – The word “post” in post-weaning diarrhea appears two times instead of one.

Thanks for suggesting on this. The sentence was changed.

Line 74 – Reference 16 links to an article in bibliography that seems not to deal with information reported in the sentence it is linked to.

Thanks for the comment, we have removed the reference.

Line 76 – Adequate references should be added to the statement.

Thanks for the suggestion, we added the reference Huang et al. (2018) [16].

Materials and Methods

Line 84 – Besides chemical composition, since this manuscript widely refers to tannins, it would be interesting to provide data regarding the composition in tannins of these extracts and their abundance. This could also help in understanding the different behaviours exerted by the two extracts, both before and after digestion.

Thanks for suggesting on this. The extracts were analysis for the chemical composition (Ashes, Crude protein, ether extract, NDF). Chemical analysis, included in table 1, have been places in the material and method section.

Line 114 – The method of expression the AOX results should be slightly clarified.

Thanks for the comment. Method of expression of AOX results was clarified as the percentage inhibition of absorbance at 734 nm was calculated and plotted as a function of the concentration of trolox standard. AOX results are expressed as µmol Trolox equivalents (TE)/ g extract of tannin.

Line 133 – It is not clear if provided informations are referred to the measurement of bacterial growth with tannins or to the calibration curve, which should be explained more extensively. More generally, this entire paragraph should be re-written a little more clearly.

Thanks for the comment. All data obtained (OD values) have been log-transformed before statistical analysis.

To transform OD values in CFU/ml we used a calibration curve obtained monitoring the E. coli F4+ and F18+ growth over time in the same experimental conditions. At regular intervals (60 minutes), we monitored the OD and, in parallel we estimated the colony forming unit (CFU) by classic plate counting method. The data were analyzed in Excel and based on the linear equation Y = mx + b, we determined the slope and the intercept and the coefficient r^2. (data not showed).

Line 136 – The entire paragraph regarding MIC tests should be completely re-written and better explained. The scheme for the preparation of stock solution dilutions and the plates is not clear and all the assessed concentrations are not reported. Moreover, it is not clear why 200µL of stock solutions were added to the first wells of each column (already containing 100µL BHI?), a volume of stock solution that appears to be too much high if compared to the BHI volume. In addition, the incubation time of the plates before absorbance reading is not reported. Finally, giving the method used, MIC should be considered as the lowest concentration of a substance that, in this case, gives null absorbance.

Thanks for the comment. The paragraph was revised. Details related to the MIC evaluation were added in order to explain better the procedure.

Minimum inhibitory concentrations were determined in 96-well microplates by preparing solutions of tannins at increasing concentrations (ranging from 10 mg/ml to 0 mg/ml) starting form stock solution (20 mg/ml). The dilution procedure resulted in a gradient of Ch100%, Qu100%, Qu75%–Ch25%, Qu50%-Ch50% and Qu25%–Ch75% concentration from 0 to 10 mg/ml across the plate. Briefly, all wells were filled with LB broth (100 µl). Two hundred microliters of the tannin solutions were added to each well. 10 µl of a E. coli culture (approximately 106 CFU/ml) were inoculated in each well of the plate, except for the blank wells, and incubated at 37°C for 18 hours. Bacterial growth was determined by the change in absorbance after reading the microplates at 600 nm in a spectrophotometer reader (BioRAD). The MIC was defined as the lowest tannin concentration that did not produce turbidity by comparison with tannin-free control (0 mg/ml) [21].

Line 153 – The purpose of reporting how to calculate the percentage of bactericidal effect is not clear, because it is not shown in results.

Thanks for the comment. The percentage of bactericidal effect of every concentration tested was calculated but not showed. We preferred to show only MBC (reduction such as 99%) Table 2.

Line 161 – The starting material of in vitro digestion should be clarified: is it the starting product or the one already dissolved in water?

The starting material for in vitro digestion was 20 g of each tannin powder sample (starting product) and not the samples dissolved in water, as for other experiments performed. The concept was clarified in the revised version of the manuscript and now is “Briefly, 20 g of each tannins powder (Ch100%, Qu100%, and Qu50%-Ch50%) was mixed with...”

Line 182 – Adequate references should be added to sentences introducing IPEC-J2 cells.

Thanks for the suggestion, a recognised reference introducing the cell model was added. (Vergauwen, H. (2015). The IPEC-J2 cell line. In the Impact of Food Bioactives on Health (pp. 125-134). Springer, Cham.)

Lines 186-188 – Cultivation conditions should only be stated once.

Thanks. It has been changed accordingly.

Line 191 – “dose dose-response” should be corrected. Moreover, in this paragraph, assessed concentrations are not reported. Moreover, treatment conditions are not reported. Furthermore, parameters measured (vitality) and the test used to create the “dose-response curve” are not clarified.

Thanks for the suggestion. Dose-response was corrected and the paragraph now reported all missing info underlined by reviewer.

The novel paragraph is the further: Further, dose-response curves (cell viability) of in vitro digested Ch, Qu and Qu50%-Ch50% have been performed on IPEC-J2 cells based on antimicrobial concentrations (0-1200 µg/ml) obtained in previous experiments. Cell viability was determined after 3 hours tannin digesta treatment by MTT test as reported by manufacturer instructions.

Line 194 – “in a second the second” should be corrected. Moreover, in this second set of experiments, it is not clear which parameter is assessed and by means of which test.

Thanks for suggesting this. it has been changed.

Line 196 – The hydrogen peroxide concentration of 1 mM differs from the 0.5 mM reported in the results.

Thanks. The concentration assessed was 0.5mM, it has been modified accordingly.

Results

Line 213 – Unit of measurement for the provided data should be corrected.

It has been corrected.

Figure 1 – In the Y axis, correct umol with µmol. Moreover, the Y axis legend should correspond to what is reported in the text.

Thank you for the suggestion. It has been corrected all along the manuscript.

Line 227 – Correct ug with µg.

Thanks. It has been corrected

Figure 2 – The quality of the image provided is too low, so it is very difficult to analyse data. Moreover, authors should find a clearer way to present these data.

Thanks for the comment. Figure 2 was moved to supplementary material section with high resolution.

Lines 229-240 – The presented differences between groups seem to be very small and limited to the same logarithm for each group. These data cannot support – at those tested concentrations – an antimicrobial activity of both extracts. Even if statistical difference is shown, an antimicrobial property should be demonstrated by a complete lack of growth of the bacterium, while these data not only demonstrate the capacity of the microorganisms to grow with the addition of tannin extracts, but also to reach, at the end of the experiment, the same log10(CFU/mL) of the controls. Finally, if the MIC is found at 6000-7000 mg/L, it should not be possible to observe an antimicrobial effect at lower doses. All these data should be completely reconsidered and re-interpreted, if not removed until further analysis.

Thanks for the comment. According to reviewer suggestion, we have removed the dose response curve from the main text and moved in supplementary materials. The authors wish to show the dose response curves in Figure 2s because it can provide info on preliminary experiments on E. coli growth inhibition (not antimicrobial activity) to demonstrate the differential activity of Ch and Que in our experimental conditions. According to reviewer suggestion we also change the paragraph title into “E. coli growth inhibitory activity “. The latter statement was also used all along the revised manuscript.

The E. coli growth inhibition induced by tannins treatment at 1200 µg/ml was maintained until 6 hours incubation. After 6 hours we observed a recovery of E. coli growth which can be due to the depletion of tannin in the medium (which have been used by E. coli). MIC was calculated after 18 hours (common time point to calculate MIC in previous studies) tannin's treatment and therefore we applied higher dosages to calculate such inhibitory concentration. 

Line 243 – The incipit “In F4+ and F18+” seems to introduce readers to comments regarding Figure 3, which are not reported.

Thanks. It has been changed.

Figure 3 – As reported for data shown in figure 2, these data do not support an antimicrobial activity of these compound at 1200 mg/L, even if some little statistically significant differences are reported. In fact, in the end, the addition of extracts and their combination brings all experimental groups to the same log10(CFU/mL). All these data should be completely reconsidered and re-interpreted, if not removed until further analysis.

Thanks for the comment. The statement antimicrobial activity was changed into E. coli growth inhibitory activity and the discussion of data was reconsidered accordingly.

Line 249 – MIC and MBC results should be reported in a new paragraph with a separate title, as happens for materials and methods explanations.

Thanks. A separate paragraph was added to the results section with MIC and MBC data.

Lines 270-272 – Spacing between symbols, letters and numbers should be reviewed in this paragraph. Moreover, as happens in figure 1, all data should be presented in a graph (with statistical analysis) and appropriate comparison of AOX properties between digested and undigested extracts should be added.

Thanks for the comment. We have added a graph with aox data.

Figure 4 - As reported for data shown in figure 1 and 2, these data do not support an antimicrobial activity of these digested compounds at 1200 mg/L, even if some little statistically significant differences are reported. In fact, in the end, the addition of extracts and their combination brings all experimental groups to the same log10(CFU/mL). Moreover, an antimicrobial activity should not only reduce bacterial growth rate, but totally prevent it. All these data should be completely reconsidered and re-interpreted, if not removed until further analysis.

Thanks for the comment. The statement antimicrobial activity was changed into E. coli growth inhibitory activity and the discussion of data was reconsidered accordingly.

Figure 5 – “Different superscript letter” in caption should be corrected appropriately; caption should also report n (experimental replicates). Authors should revise comments, since it is reported that the 400µg/mL dose does not alter viability, even if a statistically significant reduction is shown in the graph for some groups. Moreover, authors should also comment why the 100µg/mL dose of Che and Qu extracts do not statistically significant increase vitality, while 50 and 200µg/mL result in a significant increase. Furthermore, in the comments, a “CTR DMEM” group is mentioned, but not clearly shown in the graph. Finally, statistical comparison between groups is not clear, thus it should be improved.

Thanks for the comment. Statistical analysis was discussed and n of replicates added. CTR DMEM was removed as it represents the 0 µg/ml bar.

The authors changed the discussion concerning effect of 400 /50 and 200µg/mL.

Lines 305-309 – This paragraph is confusing because it explains the subsequent test in two different ways, so it is not clear if cells were previously treated and then challenged, or if the challenge occurred before the treatment.

The paragraph was changed, according to reviewer suggestion into “Furthermore, we have tested the trophic effect of Qu50%-Ch50% on IPEC-J2 cells previously stressed with hydrogen peroxide or DSS. In particular, IPEC-J2 cells were pre-treated for 3 hours with in vitro digested Qu50%-Ch50% at different concentrations (1200-50 µg/ml) and further challenged with hydrogen peroxide for 1 hour or with DSS for 24 hours, to simulate in vitro a condition of oxidative and inflammatory stress at the intestinal cell epithelium level”

Figure 6 – Caption should contain n (experimental replicates). While interesting effects are clearly proved with the DSS challenge, no statistically significant difference is shown between the “DMEM” and “0.5mM H2O2” groups, so it seems that the challenge could not significantly alter cell viability. For this reason, all the other results cannot be considered as a mitigation of the oxidative stress induced by H2O2. Moreover, authors should also comment why the same Qu50-Che50 levels (1200-800-600µg/mL) that improve cell viability over 100% in the H2O2 challenge, in Figure 5 exerted a significant reduction in cell viability.

Thanks for the comment. N of replicates was added in the figure legend. Concerning H2O2 stress, the concentration of 0.5mM induced a mild condition of stress (77% cell viability over control) we have indicated and discussed a condition of mild stress into the results section. we also added the statistical difference between control and 0.5mM h2O2 which was missed in the original version.

The latter point raised by the reviewer remains to be determine. we can speculate that the combined treatment with tannins and h2o2 versus tannins and dss showed a different behaviour. This point was addressed in the discussion section.

Lines 315 – 321 – Authors should comment on the capacity of a vitality test to give information about a reduction of oxidative stress.

The cell viability test in our experimental conditions provide information on cell viability after oxidative stress induction by hydrogen peroxide. It does not provide direct results on mitigation of oxidative stress. The indicated statements have been modified accordingly.

Discussion

Lines 328-329 – This statement is not correct, because Ch had a significant lower antioxidant capacity when compared to Qu25-Ch75 (Figure 1).

Thanks, it has been changed.

Line 334 – Adequate reference should be provided.

Thanks. This is a speculation of the authors. However, if the reviewer find criticism, we can remove the sentence.

Lines 341-359 – Discussion should be revised considering that antimicrobial data of growth experiments with doses equal to or lower than 1200 mg/L do not prove an antimicrobial activity of the two extracts or mixtures.

The discussion was revised accordingly. In general, antimicrobial analysis was changed into e. coli growth inhibition. The E. coli growth inhibition induced by tannins treatment at 1200 µg/ml was maintained until 6 hours incubation. After 6 hours we observed a recovery of E. coli growth which can be due to the depletion of tannin in the medium (which have been used by E. coli). MIC was calculated after 18 hours (common time point to calculate MIC in previous studies) tannin's treatment and therefore we applied higher dosages to calculate such inhibitory concentration. 

Line 381 – The [26] reference seems to link to a wrong article.

Thanks, it has been changed with the correct reference.

Lines 383-385 – Reference should be provided.

Thanks, it has been added. This sentence is referred to Brus et al. (2013) [26].

Lines 391-396 – The two sentences tell. the same concept, so it should be condensed in only one sentence.

Thanks, a sentence was removed. Throughout the discussion, an explanation of which compounds could be responsible for the antimicrobial effect is missing, and it is not clear if the same antioxidant molecules in the extract could be considered responsible as well of the antimicrobial activity.

Conclusions

Conclusion should be properly adjusted considering that antimicrobial data of growth experiments with doses equal to or lower than 1200 mg/L do not prove an antimicrobial activity of the two extracts or mixtures.

Thanks, it has been revised accordingly.

Supplementary material

Line 419 – It is not clear if primers were chosen from sequences published in other articles. If so, adequate references should be provided for each primer.

Thanks, the primers were designed by our research group and listed in the Table S1.

Table S2 – Melting temperature for each primer should be provided.

Thank you for the suggestion, the melting temperatures have been added for each primer (Table S1).

Abstract and summary

Abstract and summary should be revised considering all the comments made to the article text.

Thanks. Both summary and abstract have been revised accordingly.

Round 2

Reviewer 1 Report

The revised version has been impoved.

Author Response

Dear reviewer, many thanks for revising the manusript again.

kind regards

Reviewer 3 Report

First of all, thank you authors for having addressed all the suggestions presented in the major revision, and having extensively commented on all of them. 

The manuscript is now written in a more correct way, many typos or smaller mistakes have been revised and some statements refined.

Although many improvements have been made, I would kindly ask you some other minor clarifications on a few points.

Summary

In the summary, it is still stated that the two extracts have an antimicrobial effect; however, having accepted the suggestions, the two extracts only have a growth-rate reducing ability on E. coli, because - at the end of all the experiments - E. coli growth still occurs if compared to the starting point (from a log10(CFU/mL) of 7 to a log10(CFU/mL) of 9). 

Conclusion

In the conclusion, it would be preferred to stress out a little bit less the antioxidant capacity of the extracts on IPEC-J2 cells, since the unique parameter presented is a vitality test (MTT assay), which does not provide data on the oxidation state of cells, but only on their vitality thanks to their metabolic activity.

MIC protocol

As regards the protocol used to perform the MIC assay, it still remains unclear the precise way to prepare dilutions in the plate. From your explanation, I understand that you've previously made dilutions of the stock solutions in water, than you've added 200uL of these dilutions in 100uL of LB. This is quite confusing, so I would kindly ask you to better explain the way you have performed this assay.

Tannins in extracts

As I addressed in the last review, besides chemical composition, informations should be provided about the composition in tannins and polyphenols of the two extracts used in this study. This could help in understanding the different behaviours exerted by the two extracts, both before and after digestion. If these data are available, they should be added.

Author Response

First of all, thank you authors for having addressed all the suggestions presented in the major revision, and having extensively commented on all of them. 

The manuscript is now written in a more correct way, many typos or smaller mistakes have been revised and some statements refined.

Although many improvements have been made, I would kindly ask you some other minor clarifications on a few points.

Dear reviewer, many thanks for your comments and suggestions that have profoundly improved the quality of the manuscript. Please find below a point-by-point response to reviewer comments and suggestions.

Summary

In the summary, it is still stated that the two extracts have an antimicrobial effect; however, having accepted the suggestions, the two extracts only have a growth-rate reducing ability on E. coli, because - at the end of all the experiments - E. coli growth still occurs if compared to the starting point (from a log10(CFU/mL) of 7 to a log10(CFU/mL) of 9). 

Thanks for the suggestion, it has been modified.

Conclusion

In the conclusion, it would be preferred to stress out a little bit less the antioxidant capacity of the extracts on IPEC-J2 cells, since the unique parameter presented is a vitality test (MTT assay), which does not provide data on the oxidation state of cells, but only on their vitality thanks to their metabolic activity.

Thanks for the suggestion; the sentence has been changed to give:” Our data demonstrate the ability of quebracho and chestnut tannin to exert antioxidant activity and E. coli growth inhibitory activity against ETEC F4+ and F18+, together with cyto-protective activity on swine intestinal epithelial cells”.

MIC protocol

As regards the protocol used to perform the MIC assay, it still remains unclear the precise way to prepare dilutions in the plate. From your explanation, I understand that you've previously made dilutions of the stock solutions in water, than you've added 200uL of these dilutions in 100uL of LB. This is quite confusing, so I would kindly ask you to better explain the way you have performed this assay.

Thanks for the comment. We have change the paragraph to give: “Minimum inhibitory concentrations were determined in 96-well microplates by preparing a gradient of tannin solutions (ranging from 10 mg/ml to 0 mg/ml). Briefly, 100 µl of the tannin solutions, 100 µl of LB broth and 10 µl of a E. coli culture (approximately 106 CFU/ml) were inoculated in each well of the plate, except for the blank wells, and incubated at 37°C for 18 hours. Bacterial growth was determined by the change in absorbance after reading the microplates at 600 nm in a spectrophotometer reader (BioRAD). The MIC was defined as the lowest tannin concentration that did not produce turbidity by comparison with tannin-free control (0 mg/ml) [21]. The experiment was repeated three times and the results were expressed as average values.”

Tannins in extracts

As I addressed in the last review, besides chemical composition, informations should be provided about the composition in tannins and polyphenols of the two extracts used in this study. This could help in understanding the different behaviours exerted by the two extracts, both before and after digestion. If these data are available, they should be added.

Thanks for the suggestion. Besides chemical composition, we have added the tannin content in the extracts which is minimum >70% in both extracts (ISO 14088). This information has now been added in the revised version of the manuscript.

This sentence has been added in the revised version of the manuscript: “Ch and Qu extracts tested in the present study were obtained by hot water solubilization and contain 75 g of tannin/100 g of dry matter (Silvateam S.p.A).”